# Self-Reported Experiences of Midwives Working in the UK across Three Phases during COVID-19: A Cross-Sectional Study

**DOI:** 10.3390/ijerph192013000

**Published:** 2022-10-11

**Authors:** Susan McGrory, Ruth D. Neill, Patricia Gillen, Paula McFadden, Jill Manthorpe, Jermaine Ravalier, John Mallett, Heike Schroder, Denise Currie, John Moriarty, Patricia Nicholl

**Affiliations:** 1School of Nursing and Paramedic Science, Magee Campus, Ulster University, Londonderry BT48 7JL, UK; 2School of Medicine, Ulster University, Londonderry BT48 7JL, UK; 3School of Nursing and Paramedic Science, Belfast Campus, Ulster University, Belfast BT15 1ED, UK; 4Southern Health and Social Care Trust, 10 Moyallen Road, Gilford BT63 5JX, UK; 5School of Applied Social Policy Sciences, Magee Campus, Ulster University, Londonderry BT48 7JL, UK; 6NIHR Policy Research Unit in Health and Social Care Workforce, King’s College London, 22 Kingsway, Holborn, London WC2B 6LE, UK; 7School of Science, Bath Spa University, Newton Park, Newton St Loe, Bath BA2 9BN, UK; 8School of Psychology, Coleraine Campus, Ulster University, Cromore Road, Coleraine BT52 1SA, UK; 9Queen’s Management School, Queen’s University Belfast, 185 Stranmillis Road, Belfast BT9 5EE, UK; 10School of Social Sciences, Education and Social Work, Queen’s University Belfast, 69-71 University Street, Belfast BT7 1HL, UK

**Keywords:** COVID-19, midwifery, maternity care, stress, burnout, workplace

## Abstract

Maternity services cannot be postponed due to the nature of this service, however, the pandemic resulted in wide-ranging and significant changes to working practices and services. This paper aims to describe UK midwives’ experiences of working during the COVID-19 pandemic. This study forms part of a larger multiple phase research project using a cross-sectional design based on an online survey. The online survey used validated psychometric tools to measure work-related quality of life, wellbeing, coping, and burnout as well as open-ended questions to further understand the experiences of staff working during the pandemic. This paper reports the qualitative data collected from the open-ended questions. The qualitative data were subjected to thematic analysis and the four main themes that emerged were ‘relentless stress/pressure’, ‘reconfiguration of services’, ‘protection of self and others’, and ‘workforce challenges’. The key conclusions were that midwives experienced a reduction in quality of working life and significant stress throughout the pandemic due to a range of factors including staffing shortages, restrictions placed on women’s partners, changes to services and management support, all of which compounded workforce pressures that existed prior to the pandemic. This research recommends consultation of front-line midwives in relation to possible changes in practice and workforce planning in preparation for crises such as a pandemic and to ensure equitable and supportive management with access to practical and psychological support.

## 1. Introduction

Following the emergence of severe respiratory syndrome coronavirus 2 (SARS-CoV-2) in Wuhan in November 2019, it quickly spread globally and was declared a pandemic by the World Health Organization (WHO) in March 2020 [1]. Whilst the pandemic, and corresponding fear and community social restrictions, impacted populations worldwide [2], health and social care workers have been at the forefront of dealing with the consequences, with negative impacts on their own wellbeing through increased stress, variable coping, and burnout [3,4,5]. Moreover, evidence suggests that health and care staff’s wellbeing has deteriorated further over the course of the pandemic [6].

There is little research evidence on the impact of the pandemic on midwives, particularly in the United Kingdom (UK). Studies in Australia, the United States (US), and the UK had highlighted the significant stress and burnout experienced by midwives before the pandemic [7,8,9]. However, a recent survey conducted by the Royal College of Midwives in the UK found that 57% of midwives or maternity support workers (MSWs) were considering leaving their posts [10]. This could be linked to the higher rates of stress, burnout, and depression during the first wave of COVID-19, as reported in a study of nurses and midwives in Turkey [11].

Stress impacts both midwives and the women they care for. The provision of woman-centered care is central to the practice of midwifery [12] and some of the restrictions and changes in practice arising from the COVID-19 pandemic have been reported to compromise this care, leading to professional stress and personal conflict [12,13,14,15]. Fear of contracting COVID-19 and the need to protect their own families have been noted to give rise to significant stress for many midwives [14,16,17,18]. In the early days of the pandemic in the UK, the lack of or restricted access to personal protective equipment (PPE) was also of major concern, particularly in community settings [18,19,20]. When available, midwives described the additional time required to don or change PPE, the discomfort of wearing PPE for long periods, and the negative impact of wearing PPE in terms of communication [18,20,21]. The variation and in some instances lack of protocols for the care of COVID-19 positive women were also a source of concern for some midwives [20,22], particularly as awareness of the increased risks in contracting COVID-19 whilst pregnant emerged [23].

A frequent service change was the reduction in face-to-face appointments, particularly in antenatal care with phone calls and virtual meetings used instead [15,18,24]. In Australia, Wilson et al. (2021) found that over half of women had experienced some form of remote care, with reduced face-to-face contact with healthcare staff leaving them feeling they were ‘doing it alone’ [25] (p. 30). Concern was also expressed by some midwives that there was an increased risk of missing important cues about women’s health and that it can be harder to build rapport online [14,15,21]. However, some midwives considered that remote care also offered positive opportunities with improved accessibility [14] and increased self-care by women [26].

Restrictions for partners attending appointments, being present in labour, and post-natal visits varied across the duration of the pandemic and differed across the UK, with the degree of restriction generally varying according to the COVID-19 status of women [18,24,27]. Such restrictions were anxiety-provoking and distressing for many women, their families, and midwives. Hearn et al. (2021) [14] described the difficulty midwives experienced in enforcing restrictions that caused some individuals great distress, and the loneliness experienced by women. Wilson et al. (2021) acknowledged that pregnancy and labour are already anxious times for many women, so restricting partners’ contact potentially added to their distress and loneliness [20,25], as well as some partners also feeling isolated and angry [24,28].

Maternity services differ from many other health services as they cannot be stood down. Therefore, an understanding of the experiences of midwives working during COVID-19 may help guide future pandemic preparation and recovery. This study aims to assist in such understanding by analysing the experiences of a sample of UK midwives working during the COVID-19 pandemic.

### Theoretical Framework

This paper was inspired by the conceptual application of the job demands–resources (JD-R) model [29], which explains how job demands and job resources interact to impact on job-related stress, burnout, and wellbeing [30]. Job demands are described as effort, both physical and psychological, required by the job, and job resources as factors that can ameliorate or ‘buffer’ job demands, such as personal growth, learning, and development [31]. It is further argued that there are underlying psychological processes that contribute to job-related strain and motivation [31]. Health impairment is the result of chronic job demands leading to exhaustion of the employee’s physical and psychological resources and potential breakdown if demands continue for an extended period. The JD-R model was expanded to include personal resources and individual characteristics such as coping, that may help to explain the difference in how individuals are affected in response to additional job demands [32], such as those experienced by health and social care staff during the COVID-19 pandemic. Other theoretical approaches, designed for quantitative analysis (for example, Siegrist’s effort reward theory [33]), did not provide the relevant structure for qualitative analysis.

## 2. Materials and Methods

### 2.1. Data and Participants

This study is part of a larger multiple phase research project entitled ‘Health and social care workers’ (HSC) quality of working life and coping while working during the COVID-19 pandemic’ [34]. The overall project utilises qualitative and quantitative methods to explore psychological wellbeing, quality of working life, coping strategies, and burnout in nurses, midwives, allied health professionals, social care workers, and social workers in the UK over time during the pandemic. Psychological wellbeing has been described as ‘the ability to maintain a sense of autonomy, self-acceptance, personal growth, purpose in life and self-esteem’ [35] (p. 2).

The wider study employs a cross-sectional design, collecting data at approximately 6-month intervals. Data for the current analysis were collected across three time points in the pandemic: Phase 1—7 May to 3 July 2020, Phase 2—17 November 2020 to 1 February 2021, and Phase 3—10 May to 2 July 2021. The research used an online survey with reliable and validated measures. It also contained a small number of open-ended questions (see Table A1, Appendix A) to further understand the self-reported experiences of the HSC workforce as they worked through the COVID-19 pandemic. The qualitative data from responses to the open-ended questions are reported in this present paper.

The survey drew on a convenience sample of HSC workers including midwives. Study recruitment was through social media platforms (Facebook, Twitter) and via professional associations, workplace unions, professional communications, employers, and regulatory bodies. Study eligibility was based on respondents self-reporting their occupation and country of work.

### 2.2. Data Analysis

The dataset was cleaned, and irrelevant data removed to only include responses from midwives across the three study phases. For quantitative demographic questions, descriptive statistics were analysed using SPSS Version 26. Qualitative responses were recorded for 381 out of 426 midwife respondents. Thematic analysis was undertaken through a process of initial familiarisation and then the data were coded for key themes. Once coded, identified themes were reviewed by Authors 1 and 2; then refined by Authors 1, 2, and 3 into broad themes and subthemes [36]. Thematic analysis is a flexible approach to qualitative data analysis that can be used for a variety of data collection methods [37]. It can be used to analyse data from qualitative survey questions, and although responses to some open-ended questions may be brief and others more detailed, when the dataset is viewed as a whole, it can provide rich analysis and understanding [36]. It has been suggested that thematic analysis should be conducted across the whole dataset rather than by individual questions [38] and this was the approach taken in this study, with initial coding carried out across the dataset. Thematic analysis of qualitative data obtained through surveys has been undertaken successfully in similar studies [39,40].

## 3. Results

### 3.1. Demographic Data

The sample across all three phases consisted of 426 midwives (Phase 1: n = 180; Phase 2: n = 75; Phase 3: n = 171). Respondents were mainly of White ethnicity (97.2%), predominately female (99.5%), a fifth had more than 30 years’ experience (23.2%), just under a third were in the 40–49 age category (30.3%). The respondents were mainly from NI (41.4%), then England (28.9%), Wales (23.7%), and the lowest response was from Scotland (6.3%). Just under two-thirds of respondents worked in hospital settings (62.9%). Most reported no disability (93.0%). A full breakdown of the demographics across the phases is reported in Table 1.

### 3.2. Thematic Analysis

Four themes and 14 subthemes were identified from the open-ended responses which are presented in Table 2.

### 3.3. Relentless Stress/Pressure

The first theme identified was relentless stress/pressure with subthemes of work practices and client related.

#### 3.3.1. Work Practices

Reports of stress by midwives became more pronounced across the three study phases. Some midwives reported a high level of stress resulting from changes to working practices which were exacerbated by changing guidelines, with some working additional hours to cover for absent colleagues or to cope with the greater workload.

At Phase 1, several respondents commented on the stress they were experiencing due to a range of factors:

*I’m a midwife working in both community and hospital environments to provide continuity of care for women. This requires me to be on call 24/36 h per week for women living in the area I cover. However, the health board I work for now wish for me to go on call for the hospital cover due to short staffing with COVID. We are expected to do excess work without excess pay. Due to staff shielding at home midwives are already covering other midwives’ caseload so having higher patient numbers that what was deemed appropriate for this model or care. I think now, midwives are starting to feel burnt out, with the rapidly changing protocols and excess work*. (Scotland, Hospital, Phase 1)

High levels of pressure and stress continued to be reported in Phase 2. Many respondents described feeling exhausted or burnt out with several identifying numerous on-going pressures in relation to staffing, workload, loss of public support, and the need to protect their own families as contributing to the high levels of stress:

*The pressure at times feels relentless, service users can often become critical and voice their opinions with staff they come in contact even though that department is not theirs. eg criticism of waiting times for ED (emergency department treatment) or for cancer treatment is not something we in maternity services can respond to other than to acknowledge that currently there is widespread pressure within the NHS*. (NI, Hospital, Phase 2)*This caused monumental stress and poor mental health as it felt like we were being used as a staff bank while juggling oversubscribed caseloads of women in rapidly evolving guidelines as well as coming into contact with the most amount of people/in and out of their homes*. (Scotland, Hospital and Community, Phase 2)

By Phase 3, the sources of stress remained similar, such as staffing problems, and increased workload:

*Increased levels of stress. Feeling scared of impact of COVID and potentially sharing it with loved ones. A change in delivery of service or minimising home visits, but as only midwifery working keeping ladies on longer than usual. Feeling extreme pressure to get everything done* (Wales, Community, Phase 3)

Respondents also commented on the impact of stress and increased pressures on their mental health:

*I had a panic attack whilst driving on the motorway on my way to work in March. I’ve not been at work since then. I’ve been diagnosed with OCD, PTSD, intrusive thoughts and anxiety* (England, Other, Phase 3)*Everyday just feels the same and there seems to be no enjoyment or things to look forward to anymore, I sometimes feel there will be no end to this situation. I have withdrawn somewhat and get quite angry and frustrated* (Scotland, Hospital, Phase 3)

Reasons for the increased stress and reduced quality of working life seemed to be the consequence of a range of factors related to other themes that emerged from the data across the three phases.

#### 3.3.2. Client Related

One of the most frequently reported sources of stress and pressure was the implementation of and impact of partner and family restrictions. Such restrictions were aimed at trying to reduce COVID-19 transmission and fatalities. However, they caused stress and anxiety for both women and midwives with many respondents noting that increasing anxiety and emotions meant women required additional professional support. Furthermore, increasing worry, higher support needs, and sometimes abuse from women and families were reported across all three study phases:

*Women not having their birth partner for all of their appointments and in the postnatal wards. Women were highly anxious and tearful*. (NI, Hospital, Phase 2)*I have been dealing with increased verbal aggression and hostility from patients and their families, due to their frustrations at poorer quality of care*. (Wales, Community, Phase 3)*No visitors, so extra help needed by New Mums* (England, Hospital, Phase 3)

### 3.4. Reconfiguration of Services

The second theme to emerge was the reconfiguration of services with associated subthemes of changes to ante/post-natal/post-care, home birth and home visits, the impact of other service closures, and additional workload.

#### 3.4.1. Ante/Post-Natal, Home Birth, Home Visit Changes

Midwives noted that most maternity services have to be delivered within a designated timeframe. Pandemic-related changes to the organisation of care, particularly antenatal and post-natal care, involved a reduction in home visits in some areas. Many other changes were made to services across the three phases, such as increased use of telephone calls to replace face-to-face contact and the closure of midwifery led units (MLUs), which were sometimes being used as isolation wards for women who were COVID-19 positive. While some respondents reported an increase in requests for a planned home birth, usually the result of women wanting to avoid hospital and the perceived risk of contracting COVID-19 whilst there, closures of home birth services were reported by several:

*Increased use of telephone triage effectively reducing the number of face-to-face triage admissions. Conversion of MLU rooms to COVID-19 isolation areas and removal of water birth availability* (England, Hospital, Phase 1)*Increase in women wishing local Midwife birth Centre and Homebirths*. (Wales, Community, Phase 1)*The MLU has been closed to deal with the virus so we have lost a lot of our services and everything operating out of the obstetric unit. This means all our women have to give birth in the obstetric unit which, for some, is really far away. Lost the home birth service as well*. (England, Hospital, Phase 1)

#### 3.4.2. Impact of Closure/Changes to Other Services

Workload was further impacted by the closure or reduced availability of non-maternity services and buildings previously used to hold maternity clinics being no longer available:

*Increased dramatically as GPs didn’t allow clinics or women into practice, areas all clinics had to be centralised into a hub* (NI, Hospital, Phase 2)

#### 3.4.3. Additional Workload

Midwives’ workload during Phase 2 seemed strongly affected by the availability of other services. Relocation and amalgamation of services were reported by several respondents as part of coping with the pandemic’s impacts:

*Increased workload as the low risk unit MLU closed, to be used for suspected and positive COVID patients. With high risk staff being redeployed the skill mix isn’t great*. (NI, Hospital, Phase 1)*Service provision moved out of GP surgeries into Health Centres we have moved 3 times into different centres since the pandemic began. The last move was as a consequence of the rooms becoming a hub for COVID vaccinations. Some elements of care are now carried out on the phone where staff are able to work from home once a week or whilst isolating if well* (England, Community, Phase 2)

Some of the changes to working practices and services, such as swabbing for COVID-19 and the additional administration this involved, further increased the workload pressures for some midwives:

*A&E closed as a result we had to take on early pregnancy. More inductions and* C/S (Caesarean sections) *from neighbouring hospital (same Trust). High volume of phone calls booking patients in for induction or C/S from antenatal clinics from both sites not to mention our own workload in the admission and assessment unit, and only 1 midwife and maternity support worker, a receptionist would be handy! By the time we put apron and gloves on the phone rings non stop and we have to wash hands and go and answer the phone several times while dealing with patients. Also extra time required for COVID swabbing, it has been challenging and exhausting*. (NI, Hospital, Phase 1)*More admin with visiting questionnaires, contact tracing and arranging visiting time slots. This has been an expected midwifery staff role!* (Scotland, Hospital, Phase 3)

The numbers of COVID-19 positive pregnant women were reported to be increasing across the phases and resulted in additional pressures for finding suitable space for isolation and for the donning/doffing of PPE. By Phase 3, some noted the increased complexity of care required by women who were testing positive for COVID-19:

*I have seen significantly increased numbers of COVID positive women since July, compared to back in March to July. Which is very stressful & causes the worry of possibly bringing the illness home to family. PPE not great quality—worry it’s not fit for purpose*. (NI, Hospital, Phase 2)*Increased demand for Complex midwifery care for women with COVID. Increased pressure on staff due to midwives being allocated to care for COVID women, leaving the rest of the ward understaffed*. (England, Hospital, Phase 3)

### 3.5. Protection of Self and Others (Physical and Emotional/Psychological)

The protection of self and family emerged as a theme, with respondents commenting particularly on PPE, the need to protect family members, and the challenge of managing their home/work life balance, sometimes to the detriment of themselves and their families.

#### 3.5.1. PPE

During Phase 1, the provision of adequate PPE was a substantial concern for many midwives while some also mentioned the need for more timely availability of COVID-19 tests when required. By Phases 2–3, more commented on the time involved in donning/doffing PPE and its impact on their ability to communicate with women:

*When I raised a concern that we were not allowed to wear an ffp3 mask when caring for a labouring woman with suspected COVID I felt that I was in very close proximity and for an extended amount of time with her breathing Entonox I felt I wasn’t protected when I was only wearing a surgical mask. I spoke to my line manager who then spoke to the risk assessment manager and it was agreed we are now allowed an ffp3 mask but only for labour. This was very stressful for me but the end result was good*. (NI, Hospital, Phase 1)*Advice around PPE was inadequate in the few weeks and this advice changed frequently. It was felt that the advice depended on the available supply of PPE!* (Wales, Other, Phase 1)*Communication was more difficult due to masks. All just added to level of stress as at the beginning it was not known how COVID was transmitted and I had to have a student midwife with me on community who could not drive and was using public transport at the time. We had no advice at the time except we were to wear masks all the time and drive with the windows down. This was not always practical if going on longer distances and higher speeds* (Scotland, GP Based, Phase 3)

#### 3.5.2. Home/Work Balance

Midwives expressed concern about the impact of their job on their families and their ability to manage home responsibilities during the pandemic. Many found their caring responsibilities increased as they needed to protect vulnerable family members. The pandemic highlighted the difficulties in maintaining a satisfactory work/life balance for many respondents. Changes and added pressure at work needed to be managed alongside stress outside work caused by national and local lockdowns, home schooling, and lack of childcare. For some, the boundaries between work and home life became increasingly blurred:

*I find it difficult to separate the two and often have to complete e-learning at home. Non work responsibilities seem to take a back seat as I am so tired* (Wales, Community, Phase 2)*Work has overtaken home—I am behind on all home ‘stuff’ and on the back foot with family plans. Not enough brain space to be in control of it* (England, Hospital, Phase 3)*Non-work—difficult to be school teacher on top of all the work that we had to do and deal with all the stress and pressure. Felt I had little time for parenting. No outlet to go out with friends or family to celebrate birthdays or occasions and this is a huge loss to life*. (NI, Hospital, Phase 3)

#### 3.5.3. Protection of Family

A desire to protect family emerged strongly from the data and remained across the three phases, with midwives adopting various strategies to reduce the risks of their job affecting others at home:

*Husband has become sole carer for my children as I have often had to work longer hours and overtime shifts. My mum has been shielding as she is elderly and due to me working in the NHS she has had to depend on meals on wheels etc and my daughter helping her with shopping etc as I am staying away from her in order to protect her* (Wales, Hospital, Phase 1)*My Husband has been shielding, I have had to live in a separate part of the house so I have had to change clothing in the car, clean down everything daily, wash all things delivered to the house. It has meant I stay at work longer to finish things I could have done at home* (England, Community, Phase 3)

### 3.6. Workforce Challenges

The final theme was workforce challenges, and included subthemes of staffing, management and leadership, communication, inequity, and feeling valued.

#### 3.6.1. Staffing

Unsurprisingly, staffing was a major concern for many respondents in Phase 1. They reported a reduction in staff numbers due to colleagues shielding, isolating, and being absent due to contracting COVID-19. On occasion, this contributed to changes in staff ratios and skill mix which have been shown to have a negative impact on the safe delivery of care [41]. These problems continued over Phases 2 and 3, with some midwives describing heightened stress due to continually having to cover extra shifts at short notice:

*Due to staff shielding at home midwives are already covering other midwives’ caseload so having higher patient numbers than what was deemed appropriate for this model or care. I think now, midwives are starting to feel burnt out, with the rapidly changing protocols and excess work*. (Scotland, Hospital, Phase 1)*Significant staff shortages due to sickness or shielding, doubled caseload due to less midwives, changing ways of working, more use of technology, increased anxiety/dissatisfaction from some service users*. (Wales, Community, Phase 3)

#### 3.6.2. Management and Leadership

Respondents identified management practices that had worked well during Phases 1 and 2 and areas that could have been improved in terms of management support. Communication was identified as a vital part of management support and, where this worked well, staff felt that they were listened to, that their manager was visible and ‘present’, and that they were kept up to date with clear guidelines. The visibility of management was important to some as this indicated to them that managers understood the clinical pressures being experienced by midwives:

*The most important thing was INFORMATION! Excellent daily communication from the CEO (Trust Chief Executive) kept us all in the loop!* (England, Hospital, Phase 1)*Having an excellent Manager who I can turn to for advice and excellent colleagues who all support each other.* (NI, Hospital, Phase 2)

#### 3.6.3. Communication

Where improvements were suggested, the same areas as described above were identified, including the need for visible managers who used effective communication at both organisational and local levels:

*Clearer messages in the early days—conflicting messages on PPE from professional bodies, Department of Health etc. Once the PHE (Public Health England) advice came out everything was much clearer*. (NI, Hospital Phase 1)*Ask the staff what works well as well as reading the statistics. Staff are doing over and above what is expected of them to get the best outcomes for mums and babies. Perhaps it would make staff feel their opinion is valued*. (Scotland, Hospital, Phase 1)*Managers to clinically work at point of care to see pressure of work, as most managers not clinical anymore. Work pressure and lack of resource, and staff substantially increased over last years.* (NI, Hospital, Phase 2)

#### 3.6.4. Inequity

Feelings of inequity were evident in some responses, ranging from racial discrimination to disparity in the way staff were treated by management and concerns about certain terms and conditions:

*Dealing with the blatant racism and discrimination in the workplace* (England, Community, Phase 1)*Terms and conditions weren’t fair. Staff who were shielded got to keep annual leave and those who worked throughout got to keep none!!!! This was demoralising and caused animosity amongst staff* (NI, Hospital, Phase 2)*We are left to get on with it. Management tell us don’t do this etc. Yet they (are) working from home. Condensed hours and not visible when we need them. Want to keep midwifery led “open at all costs” to detriment of high risk patients and staff in those areas. Workload disproportionately allocated.* (NI, Hospital, Phase 2)*Haven’t felt at all supported by my employer during the pandemic. I enjoy working with patients/service users but struggle with an oppressive, undermining culture that has become more evident during the pandemic.* (NI, Hospital, Phase 3)

#### 3.6.5. Feeling Valued

The need for managers to thank staff for their efforts during the pandemic was mentioned across all phases. Many respondents also commented on the high level of public support in Phase 1. The provisions of free parking and sometimes staff meals were also seen as important signs of appreciation:

*Colleagues have been inspirational, newsletters frequently and emails of praise and thanks from management* (NI, Hospital, Phase 1)*Emails from chief executive saying that appreciated the work as we’re doing. Facebook campaign to recognise unsung heroes, good for morale, we are all in it together.* (NI, Community, Phase 1)

However, by Phases 2 and 3, respondents were mentioning a perceived waning of public support, and as described earlier in relation to partner restrictions, an increase in verbal abuse from women and their visitors, which affected morale:

*At the start the public showed appreciation of the care provided but as time has gone on it feels it’s always give and more is expected of you. And despite being kind, well mannered and wanting the best for those you provide care for, the relentless pressure and increase workload it has become frustrating*. (NI, Hospital, Phase 2)

## 4. Discussion

The findings of this study provide illustrations of the considerable pressure experienced by midwives in the UK during the first year of the pandemic (May 2020–May 2021).

In terms of the job demands–resource model, the findings from this study suggest that midwives experienced a considerable increase in job demands during the pandemic due to a range of factors including staffing shortages, changes to services, and partner restrictions, in addition to the service pressures that were evident prior to the pandemic [7] (see Figure 1).

Across the phases examined (May 2020–May 2021), workforce challenges emerged strongly as a major source of stress with low staffing levels compounded by the number of staff absent due to COVID-19-related reasons, sickness, shielding, or isolating. This, together with the additional workload caused by increased infection control measures (cleaning, administration, testing, donning and doffing PPE), providing additional support to anxious women due to partner restrictions, and extra work covering for closures/suspensions of other services, resulted in higher job demands for midwives. These findings bear similarities to those from other countries including Indonesia [18], New Zealand [19], and Australia [21] where staffing proved to be a significant challenge. Whilst it is to be expected that some staffing pressures will be relieved as the pandemic eases and infection rates reduce, there is an existing shortfall of 2000 midwives in England [10], and there has been a fall of 300 in the numbers of midwives over the previous two months to July 2021 in England [42]. A review of maternity services in one NHS Trust in England highlighted the need for significant investment in ensuring staffing levels that can provide safe and effective care [41].

Job demands were also increased due to the need to alter services in response to a rapidly evolving situation during the pandemic, and midwives in this study described an increase in the use of remote care particularly in relation to antenatal and post-natal care through phone and video conferencing. Midwives, however, expressed concern that safety may be compromised by important information or observations being missed, a view echoed by Australian and Spanish midwives [14,43]. However, remote care can have some advantages, including widening access to antenatal education and increasing self-care by women [21,26].

This study revealed a mixed picture in relation to home births with some midwives reporting an increase in demand, whereas others described the suspension of the local home birth service. Midwives also reported the closure of MLUs in some areas with services concentrated in central, usually hospital-based hubs. The demands of changing services further contributed to pressure on midwives, not only in adapting to rapidly changing practices but also dealing with the frustration and anxiety of some women denied a service such as home birth when this was apparently still available in other areas.

Restrictions to partners accompanying and visiting women emerged as a difficult area, supporting findings from other studies [14,24]. Further demand was placed on midwives in supporting increasingly anxious women who were missing their partners. Bradfield and colleagues have noted that these restrictions caused frustration for some women and families, resulting in increased anger and abuse directed towards midwives [24]. In the UK, this was further exacerbated by the variations in restrictions placed on partners and families across the regions, further adding to the stress and confusion of women and midwives [27]. Whilst it is recognised that many difficult and speedy decisions were required due to the rapid spread of COVID-19, and that devolution permits variations across the UK, greater pandemic preparedness might assist front line staff with better information about why such variations exist and how to communicate them.

The need to protect self and family emerged as a concern for midwives as in studies elsewhere [13,18,19,43]. In Phase 1 of our study, the availability of PPE was a major concern but over time this receded, and comments centred around additional time required to use PPE and the negative impact on communication with women [14,18,21,22]. Across Phases 2 and 3, concerns regarding self-protection problems were related more to maintaining a manageable work/home life balance, and the resulting impact on their own and their family members’ wellbeing. A similar finding was evident in a study of midwives in Spain, where participants experienced increased home responsibilities and the need to better balance personal- and work-related responsibilities [43]. The availability of more flexible working arrangements could have a mitigating effect, and, where these were available, some midwives responded very positively, indeed they expressed concern that these arrangements might end as the pandemic eased.

Good leadership and management have the potential to increase job resources available to staff [44] and it has been suggested that as stress increases with rising job demands, then the provision of stable job resources becomes vital to prevent burnout [45]. Managers’ actions had an impact on midwives’ experiences during the pandemic, with the most important factors including communication, managers being ‘visible’, providing clear guidelines, and both practical and psychological support for their staff. Where midwives reported positive experiences, they identified that they felt listened to and valued by managers, and that managers were both visible and supportive within clinical areas. Conversely, where the experience had been negative, suggestions for improvements covered these same areas. The findings of this present study also highlighted the need for both practical (flexible working, adequate rest facilities) as well as emotional support (receiving thanks, mental health support services, staff ‘wobble’ rooms). A report by the Society of Occupational Medicine [46] examining the wellbeing of nurses and midwives in the UK stressed the need for stakeholders to work together to ensure optimum staffing, a properly resourced and implemented occupational mental health strategy, suitable facilities for breaks, and warned of the risk to safe and effective care for patients if staff wellbeing is not adequately considered.

Midwives in this present study identified ‘being listened to’ as an important factor in positive management during the pandemic. Frustrations were expressed over the constant changes in guidelines and restrictions with midwives feeling they had no control over these uncertainties. In Australia, midwives expressed similar frustrations, reporting the inconsistencies between limiting visits to less than 15 min whilst they spent up to 12 h with a woman in labour [15]; likewise, in Spain inadequate guidelines and protocols caused increased anxiety and fear [43]. The apparent lack of influence on such changes and the impact that restrictions and service adaptions had on midwives’ ability to provide quality woman-centred care and ensure safety were sources of anxiety and distress for some midwives [14,43]. Walton (2020) suggested that making changes within midwifery services may prove beneficial not just during the pandemic but for the future [47]. To do this, midwives must be consulted, respond proactively, and engage in disaster or pandemic planning. As underlined by Eagen-Torkko et al. (2021), midwives have their own perspectives and have an ethical obligation to speak up when they can see policies which may have unintended harms [12]. This will necessitate greater engagement with local and national pandemic preparedness.

The importance of being valued by both management and the wider public was evident across the phases, with an increase in abuse directed at midwives noted here and more widely [21]. A Royal College of Midwives’ (RCM) survey reported similar findings with 92% of midwives suggesting they are not valued by the Government, 54% feeling not valued by their employer, and only 47% feeling valued by the general public, although 82% did feel that women and their families valued them [10]. Such feelings may reflect more fundamental problems than those related to the pandemic. Job resources including appreciation have been found to be supportive for staff engagement when job demands are high [44]. A further concern highlighted within this study was inequity in the workplace including discrimination, unfair workload allocation, and favouritism as well as unfavourable terms and conditions. Concerns have been raised for over the 20 years about a bullying culture within midwifery [48,49]. The Ockenden Report (2022) described significant culture problems in the midwifery profession, with staff afraid to speak out and evidence of bullying, with 65.5% of its survey respondents noting that midwives had either witnessed or experienced bullying in the workplace [41]. This lack of psychological safety is known to impact negatively on the safety of care provision [41]. Again, such problems pre-date the pandemic but have been brought into sharp relief by its pressures.

The significantly higher rates of burnout reported in this wider study for midwives compared with other health and social care workers [34] have been corroborated by studies in Turkey which examined nurses and midwives [11,50]. While the reasons behind this remain unclear, research conducted pre-pandemic has highlighted that work-related stress and burnout already existed within the midwifery profession and were both higher than in the general population, and particularly high in the UK in comparison with other countries [7,9].

The findings of this study are important in relation to both the wellbeing of midwives and the quality and safety of care provided to women and their babies. Outcomes in maternity care were found to have worsened due to the pandemic, with increased maternal deaths, stillbirths, and maternal depression [51,52]. Furthermore, studies of respectful maternity care during the pandemic found that changes in services and restrictions to social support had led to compromises in quality of care [53,54]. It is argued that increased staffing in maternity care can reduce incidence of post-partum haemorrhage, maternal readmission, neonate resuscitation, and epidural use [55], potentially increasing safety. Staffing issues were identified as a significant issue for midwives in our study and addressing this through provision of improved job resources could improve staff wellbeing [56], and improve retention and therefore staffing levels and quality and safety of care [41,57].

This present study illustrates the increase in job demands experienced by midwives in the UK during the pandemic. Some midwives reported the positive impact that leadership and management could have through provision of practical and emotional support, including good communication, clear guidelines and acknowledgement of midwives’ value, resources that have the potential to ‘buffer’ the impact of increased job demands. However, other midwives described the impact of a lack of support from management, poor communication, and inequity that may serve to compound the increase in job demands, resulting in decreased health and wellbeing. Midwives reported a perceived increase in stress, and some the significant negative impact on their mental health. This highlights the need to acknowledge the high demands placed on staff and increased stress experienced, putting in place practical and psychological supports aimed at ensuring the job demands and job/personal resources are more effectively balanced.

### 4.1. Recommendations

There is a need to ensure that midwives have access to adequate job resources in terms of more flexible working arrangements, appropriate development opportunities, good leadership, and support. Workforce planning needs to take account of the impact of the pandemic which has exacerbated pre-existing problems evident in maternity care in order to ensure required staffing levels can be achieved and maintained. Safety for women and their babies can be optimised by actively engaging midwives in the planning and design of maternity service provision underpinned by appropriate and accessible multidisciplinary training and education. Future studies could explore the longitudinal impact of changing conditions on midwives’ work-related quality of life domains such as job satisfaction and wellbeing over time with as a consistent sample or cohort as possible.

### 4.2. Strengths and Limitations

A key strength of this present study is that data were collected over three time points at different stages of the pandemic, allowing for comparisons to be drawn and changes over time to be observed. It was helpful to have data from across these different time points to highlight the short-term and more medium-term working practices within the midwifery profession. This element is a strength of the study’s repeated cross-sectional design which allowed comparison and an investigation of associations between outcomes and themes [58]. However, the study is limited by the cross-sectional data collection method as this means the findings cannot be used to infer cause and effect [58].

## 5. Conclusions

In this study, we explored the experiences of UK midwives working during the COVID-19 pandemic across three time points (May–July 2020, November 2020–February 2021, May–July 2021). We found midwives faced stress and pressure due to increased job demands resulting from changes in working practices and wider contexts. This, combined with challenges in staffing levels, inequality, and lack of management support, reflects long-standing concerns within midwifery. The profession and its employers will need to work with patient groups and front line staff to ensure high quality, safe, and effective maternity services for all as we recover from the pandemic.

## Figures and Tables

**Figure 1 ijerph-19-13000-f001:**
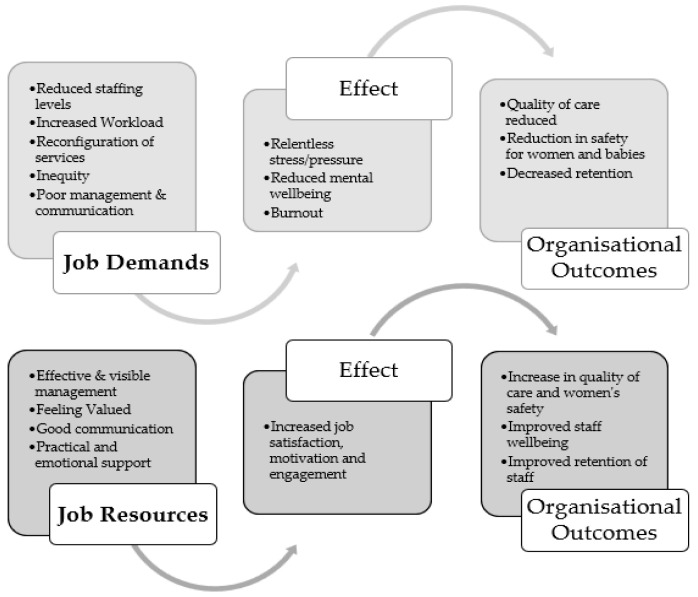
The Job Demands Resources Model.

**Table 1 ijerph-19-13000-t001:** Sociodemographic details of midwifery sample (N = 426).

Variable	Phase 1 (7 May–3 July 2020)	Phase 2 (17 November 2020–1 February 2021)	Phase 3 (10 May–2 July 2021)
Sex			
Female	180 (100%)	75 (100%)	169 (98.8%)
Age			
16–29	27 (15.0%)	5 (6.7%)	29 (17.0%)
30–39	38 (21.1%)	16 (21.3%)	45 (26.3%)
40–49	57 (31.7%)	23 (30.7%)	49 (28.7%)
50–59	44 (24.4%)	28 (37.3%)	36 (21.1%)
60–65	14 (7.8%)	3 (4.0%)	12 (7.0%)
Ethnic background			
White	174 (96.7%)	75 (100%)	165 (96.5%)
Black	2 (1.1%)	0 (0%)	3 (1.8%)
Asian	1 (0.6%)	0 (0%)	0 (0%)
Mixed	3 (1.7%)	0 (0%)	3 (1.8%)
Country of work			
England	41 (22.8%)	5 (6.7%)	77 (45.0%)
Scotland	5 (2.8%)	5 (6.7%)	17 (9.9%)
Wales	53 (29.4%)	1 (1.3%)	47 (27.5%)
Northern Ireland	81 (45.0%)	64 (85.3%)	30 (17.5%)
Number of years of work experience			
Less than 2 years	12 (6.7%)	2 (2.7%)	18 (10.6%)
2–5 years	20 (11.1%)	6 (8.0%)	29 (17.1%)
6–10 years	34 (18.9%)	8 (10.7%)	35 (20.6%)
11–20 years	39 (21.9%)	17 (22.7%)	33 (19.4%)
21–30 years	37 (20.8%)	19 (25.3%)	16 (9.4%)
More than 30 years	36 (20.2%)	23 (30.7%)	39 (22.9%)
Place of work			
Hospital	113 (62.8%)	48 (64.0%)	107 (62.6%)
Community	36 (20.0%)	17 (22.7%)	47 (27.5%)
General practice (GP) based	0 (0%)	1 (1.3%)	5 (2.9%)
Other	31 (17.2%)	9 (12.0%)	12 (7.0%)
Disability status			
Yes	12 (7.1%)	6 (8.8%)	8 (5.3%)
No	156 (92.9%)	68 (91.2%)	142 (94.0%)
Unsure	0 (0%)	0 (0%)	1 (0.7%)

Note. Presented are column percentages, which are valid percentages to account for missing data.

**Table 2 ijerph-19-13000-t002:** Qualitative analysis; theme and subthemes.

Themes	Relentless Stress/Pressure	Reconfiguration of Services	Protection of Self and Others	Workforce Challenges
Subthemes	Work practices Client related	Ante/post-natal, homebirth, home visits changes Impact of closure/changes to other services Additional workload	PPE Home/work balance Protection of family Importance of practical and emotional support	Staffing Management and leadership Communication Inequity Feeling valued data

## Data Availability

The data presented in this study are available on request from the corresponding author. The data are not publicly available due to ethical restrictions.

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
