# Peer review of "Self-Reported Experiences of Midwives Working in the UK across Three Phases during COVID-19: A Cross-Sectional Study"

_ijerph, 2022, doi:10.3390/ijerph192013000_

Round 1

Reviewer 1 Report

It is very important and valuable paper, I think. However, I need to know the reason why authors devided Covid-19 period into 3 phases. Is it just the arithmetic mean, or some condition changes appeared in each phase? If you have any reason(s), you need to describe reasons sufficiently in advance. Becase it is very interesting about comparisons among phases, if you describe the reason(s) it would increase the valueness.

You explained the modifications about PPE in the introduction. It is quite valuable to explain more detail.

Please put some definition about wellbeing (WB). Everybody says that wellbeing is important (maybe influance from SDGs), but noone said the difinition before starting mention about study on well-being. You mentioned Psychological WB, then you need to describe the difinition about it. Whose WB (ex, Ryff) you are using for?

About quantitative analysis, please descibe more details. Which statistical methods did you use for which data? State the threshold for statistical significance in materials and methods.

I think the results were mainly explained about qualitative one (except 3.3.1). Where the results of quantitative one? If you got NS for all data, you should descibe so in your paper as long as you did statistical evaluations. But I liked the results a lot. Nice challenge.

All portion of the discussion is valuable. However, you can not say like [increased] without no statistical results. Please rewrite it after consideration above comments. 

Reviewer 2 Report

This paper addresses the importance to understand maternity services during crisis i.e. the corona pandemic, and changes in demands and the effect of resources focusing on midwives experiences. The authors did a good job with choosing a timely theoretical basis (Job Demands-Resource model; JDR-M) and conducting a cross-sectional design with qualitative interviews and thematic analyses. The key results are the the four main themes which emerged. I agree that main conclusions were that midwives experienced a reduction in quality of working life (also on basis of other studies) and demands in face of the pandemic were mainly staffing shortages, limited support of partners, challenges with services and management. However, 1) the integration into the theory and 2) the results of other studies should be improved. 3) Recommendations should be expanded with regard to future research and  decision makers.

Furthermore, the authors should

4) Aggregate their findings into a figure and more clear communication in the discussion, synthesizing them in a way that readers are more likely to actually understand the data and findings, and their contribution to science and practice.

5) The whole research should be also related to patient safety as an important factor in the field, not only stipulated by the WHO...

6) In addition to the JDR-M, the authors should also consider more potential theories and models.

Round 2

Reviewer 2 Report

While I see that the authors met some of my feedback, I am not satisfied with the following:

Recommendations should be expanded with regard to future research and  decision makers. --> Add a section into the general discussion (not only the conclusions)

The whole research should be also related to patient safety as an important factor in the field, not only stipulated by the WHO. --> This needs to be extended!

Aggregate their findings into a figure and more clear communication in the discussion, synthesizing them in a way that readers are more likely to actually understand the data and findings, and their contribution to science and practice.

The table has been changed to a figure that identifies these/sub-themes more clearly. The discussion has been restructured and more clearly integrated with theory.

--> please get the old table back in and ADD a new figure which relates to the JDR and revisit my recommendation above!
